# Bio-based Catalysts from Biomass Issued after Decontamination of Effluents Rich in Copper—An Innovative Approach towards Greener Copper-based Catalysis

**Tomasz K. Olszewski [1]**, **Pauline Adler [2]** and **Claude Grison [2],***

[1]   Wroclaw University of Science and Technology, Wybrzeze Wyspianskiego 29, 50-370 Wroclaw, Poland; tomasz.olszewski@pwr.edu.pl

[2]   Laboratory of Bio-inspired Chemistry and Ecological Innovations UMR 5021 CNRS—University of Montpellier, Cap Delta, 1682 Rue de la Valsière, 34790 Grabels, France; pauline.adler@umontpellier.fr

*   Correspondence: claude.grison@cnrs.fr; Tel.: +33-4-34-35-98-71

**Abstract:** The abundance of Cu-contaminated effluents and the serious risk of contamination of the aquatic systems combine to provide strong motivating factors to tackle this environmental problem. The treatment of polluted effluents by rhizofiltration and biosorption is an interesting ecological alternative. Taking advantage of the remarkable ability of the selected plants to bioconcentrate copper into roots, these methods have been exploited for the decontamination of copper-rich effluents. Herein, we present an overview on the utility of the resulted copper-rich biomass for the preparation of novel bio-sourced copper-based catalysts for copper-mediated reactions: from the bioaccumulation of copper in plant, to the preparation and full analysis of the new Eco-Cu catalysts, and their application in selected key reactions. The hydrolysis of a thiophosphate, an Ullmann-type coupling leading to *N*- and *O*-arylated compounds, and a CuAAC "click" reaction, all performed under green and environmentally friendly conditions, will be described.

**Keywords:** rhizofiltration; biosorption; Eco-Cu catalyst; hydrolysis of thiophosphate; Ullmann-type reactions; eco-friendly *N*- and *O*-arylation; green CuAAC "click" reaction

## 1. Introduction

Numerous of the greatest successes of organometallic and inorganic chemistry are based on the use of metal complexes for catalysis and organic synthesis [1,2]. In many cases, such metal complexes, based most frequently on precious metals such as platinoids, allow the creation of new C–C or C–H bonds or the cleavage of H–H bonds, etc. Wilkinson [3,4] and, more recently, Noyori and coworkers [5] have developed remarkably reactive rhodium and ruthenium complexes for use in catalytic asymmetric hydrogenations of C=C or C=O bonds. Nowadays, many C–C, C–N, and C–O coupling reactions; C–H activations; and metal catalysed redox reactions are widely used in organic processes [6]. However, the main limitation of these catalytic systems remains the cost and availability of the metal used [6].

Therefore, the current challenges rely precisely in the use of new metal catalytic systems replacing platinoids by more abundant, cheap, and readily available metals [7]. In that scenario, copper is a transition metal that perfectly meets these criteria [8]. Indeed, the literature testifies a broad range of reactivity of copper species (Figure 1). One of the most famous is the ability of copper to promote coupling reactions [9,10], among them the Ullmann-type reactions [11–14]. Indeed, the literature offers many methodologies for the formation of C–N bonds, such as the arylation of amines [15–17], the arylation and vinylation of *N*-heterocycles [18], aromatic amidation (Goldberg reaction) [19,20],

and azidation [21]. These Ullmann-type reactions have been extended to the formation of C–O bonds that led to the synthesis of diaryl ethers [22], to the aryloxylation of vinyl halides [23,24], or to the cross-coupling of aryl halides with aliphatic alcohols [25]. Likewise, formations of C–C bonds were reported, including cross-coupling with terminal acetylene [26–29], the arylation of activated methylene compounds [30], and cyanation [31]. Worth mentioning are also the C–S bond formation (the synthesis of bisaryl- and arylalkyl-thioethers [32,33] and the assembly of aryl sulfones [34]) and the C–P bond formation by copper catalysis [35]. Another key part of copper chemistry is the 1,3-dipolar cycloaddition, in particular the Cu-catalysed Azide-Alkyne Cycloaddition (CuAAC) "click" [36,37]. Indeed, this reaction represented the prime example of the concept of "click" chemistry, developed by Sharpless et al. [38]. This reaction had a huge influence on drug discovery and became very popular in biological and medicinal science as a ligation tool [39]. Copper can also be a versatile catalyst and promote reductions (hydrosilylation of ketones [40]) or oxidations (cleavage/oxidation of benzylidene acetals [41]). As last examples, copper catalyses the protection/deprotection transformations of important functional groups such as the formation of isopropylidene acetals [42], the deprotection of thioacetals, and selective ester hydrolysis [43].

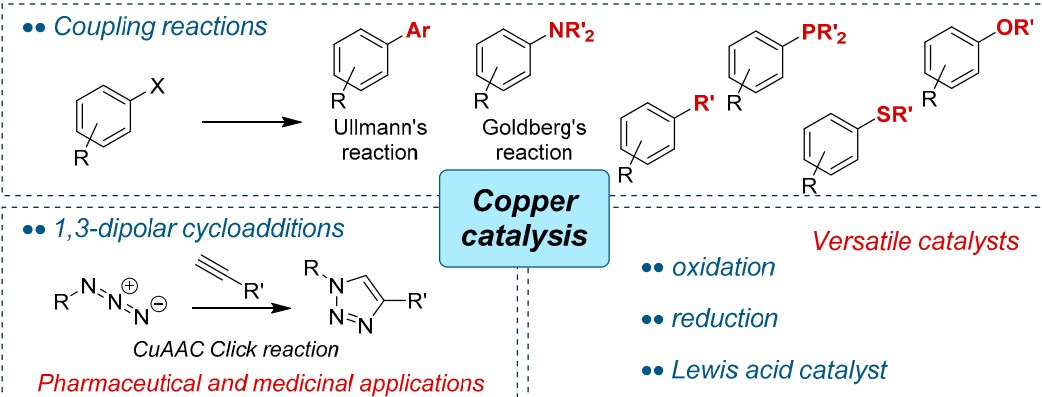

**Figure 1.** A representation of different reactivities of copper species.

Furthermore, in line with the current trends in sustainable and green chemistry, heterogenous catalysts for copper-based reactions were reported in the literature. This area of research continues to evolve as a more suitable approach because of its crucial advantages such as the easy separation of products from catalyst, a high stability of the heterogenous catalysts, and, most importantly, their recyclability. In that aspect, the selected examples of heterogenous copper catalysts employing Cu/C (copper-on-charcoal) [44,45], copper powder [46], magnetite-supported copper nanoparticles [47], Cu/ligand catalyst immobilized on silica [48], CuI immobilised on MOF [49], alumina-supported CuO [50], and recent mesoporous copper supported on manganese oxide material (meso Cu/MnOx) [51] as well as copper oxide catalysts supported on three dimensional mesoporous aluminosilicates [52] are certainly worth mentioning. Additionally, representative examples of Ullmann-type reactions using organic (bio)polymers [53–55], or zeolites [56], as supports should be mentioned.

Copper-catalysed reactions have found wide applications for the synthesis of natural products, biomolecules, and precursors of advanced materials [57–59]. Heterocycles have been efficiently prepared through copper-catalysed cascade or multicomponent reactions [60–64], some of which have led to the synthesis of pharmaceutically important compounds [65–67].

Given the importance of copper-based catalysis, the purpose of this review is to discuss the recent advances in the preparation of the first fully bio-based, ecological copper catalysts, named Eco-Cu, and their application in organic synthesis.

In a first part, this review will focus on the preparation of the ecocatalysts based on the use of Cu-rich biomasses and remedial phytotechnologies (i.e., rhizofiltration [68–70] or biosorption [71])



in order to decontaminate metal-polluted water. This part highlights the possibility and feasibility to efficiently reconcile copper-catalysis for organic synthesis with the protection of the environment, including the preservation or remediation of aquatic systems. Rhizofiltration refers to the approach in which aquatic plant roots are used to purify contaminated water through metabolically mediated process, whereas biosorption can be defined as the ability of biological materials to accumulate pollutants from wastewater through physicochemical pathways based on different mechanisms including absorption, adsorption, ion exchange, surface complexation, or precipitation.

In a second part of this review, the full characterisation of the copper ecocatalysts will be described. Polymetallic compositions, morphology, Lewis acid properties, and oxidation's states of reported ecocatalysts will be detailed in order to rationalise and predict their reactivity.

Finally, the synthetic potential of Eco-Cu will be illustrated through three major applications of copper catalysis: (i) the hydrolysis of the thiophosphate group in an important example of parathion [68], (ii) the Ullmann coupling for *N*- and *O*-arylation [69], and finally, (iii) the CuAAC "click" reaction performed under green and environmentally friendly conditions [70].

## 2. Phytoaccumulation of Copper

There are different ways to phytoaccumulate metallic elements. In this part, we will focus on the rhizofiltration and biosorption that have been used by Grison et al. to prepare copper ecocatalysts [69–71].

### 2.1. Preparation via Rhizofiltration

The rhizofiltration of copper was studied with three aquatic plants: *Bacopa monierri*, *Lolium multifolium*, and *Eichhornia crassipes* [69,70]. When choosing the plants, two criteria were selected: i) the biomass should be insoluble in water and ii) the structure of the biomass should be based on carbon-containing aromatic compounds and contain many carboxylate groups naturally present in the material of plant origin. Table 1 presents the concentration of copper in each plant after rhizofiltration and the bioconcentration factor of each accumulation. The bioconcentration rates of Cu in the roots were evaluated by ICP-MS (Inductively Coupled Plasma Mass Spectrometry).

**Table 1.** The copper concentration in roots and bioconcentration factor (BCF) calculation.

| Plant | Cu Concentration in Effluent (mg/L) | Roots (wt. % $\pm$ SD) | BCF [4] (in Roots) |
|---|---|---|---|
| *Bacopa monnieri* [1] | 10.5 | 1.34 $\pm$ 0.011 | 1279 |
| *Lolium multiflorum* [2] | 10.6 | 0.71 $\pm$ 0.0036 | 666 |
| *Eichhornia crassipes* [3] | 10.5 | 2.55 $\pm$ 0.027 | 2430 |

[1] During Cu accumulation, the substrate used for growth was removed. [2] *Lolium multiflorum* was still on Fleximix Root Riot Organic Starter Cubes during the copper accumulation. [3] No substrate was used for growth. [4] Bioconcentration factor.

The BCF is the ratio of the amount of copper accumulated in the roots to the concentration of the element left in the effluent after rhizofiltration. For all species, the initial concentrations of copper were the same. The BCF results showed that each studied aquatic plant was capable of bioconcentrating copper in large quantities. *E. crassipes*, however, was found to be the most efficient compared to *B. monnieri* and *L. multiflorum*, the latter being the least effective of the three species for this bioaccumulation. The *E. crassipes* also has the advantage of having a very important root biomass, and therefore, this plant represents the best candidate for rhizofiltration in order to prepare a large quantity of Cu-rich ecocatalysts.

### 2.2. Preparation by Biosorption

Biosorption is using natural waste, abundant and rich in tannin biomasses, as a metal accumulator. The same authors recently illustrated this methodology by using coffee grounds to accumulate different metals, in particular copper [71]. In order to increase their adsorption capacity and affinity with the

metal, the coffee grounds were functionalised with citric acid. This modification was described as easily doable and important for the efficiency of the biosorbant but also for its preservation. Table 2 depicts the biosorption of different copper solution by analysing the quantity of the element in the effluent, before and after accumulation.

**Table 2.** The biosorption of copper using functionalized coffee grounds. [1]

| Quantity of Coffee Grounds (g) | Initial Concentration of Cu (mg/L) | Final Concentration of Cu (mg/L) |
|:---:|:---:|:---:|
| 1 | 15 | 0 |
| 1 | 101 | 54 |
| 2.5 | 315 | 186 |

[1] Stirring at room temperature for 2 hours.

It appeared that the maximum biosorption capacity of the functionalised coffee grounds was very high. These data could be advantageously compared to the values reported in the literature by Cerino-Córdova et al. [72]. The authors described the accumulation of copper with closely related materials. The coffee grounds' functionalisation's conditions are different in both studies. In Cerino-Córdova's work, the esterification reaction occurred in aqueous medium, compared to Grison's conditions in ethanol. In addition, the biosorption tests were carried out at different pH, slightly acidic (pH = 5) in Cerino-Córdova's work, where a large part of the carboxylate functions was protonated, and this fact limited the biosorption of the copper. Under Grison's conditions, the biosorption was faster (2 hours compared to 5 days) at a neutral pH. The biosorption seemed then faster than the precipitation of $Cu^{2+}$ ions. The biosorption method was, therefore, ideal for recycling Cu(II) species from homogeneous catalysis reactions.

## 3. Preparation and Characterisation of the Copper Ecocatalyst, Eco-Cu

### 3.1. Preparation via Rhizofiltration

An interesting recovery of Cu-rich biomasses or Cu-rich coffee grounds is the preparation of bio-based catalysts, called Eco-Cu. A sequence of treatments was developed for the reproductible preparation of Eco-Cu [69,70]. After a controlled thermal treatment under air, the organic matter was converted into $CO_2$ and $H_2O$. The mineral residue polymetallic oxidized species were treated under acidic condition (HCl in this case) in order to form metallic chloride species. The mineral compositions of the resulting Eco-Cu were measured by ICP-MS (Inductively Coupled Plasma Mass Spectrometry) (Table 3).

**Table 3.** The mineral compositions of the different Eco-Cu.

| Eco-Cu | Plant | Mineral Composition (%wt. $\pm$ %RSD) | | | | | | | |
|:---:|:---:|:---:|:---:|:---:|:---:|:---:|:---:|:---:|:---:|
| | | Na | Mg | Al | K | Ca | Fe | Zn | Cu |
| Eco-Cu$_1$ | *Bacopa monierri* | 5.29 $\pm$0.18 | 1.86 $\pm$0.53 | 1.93 $\pm$1.14 | 3.48 $\pm$0.29 | 4.91 $\pm$0.77 | 1.21 $\pm$0.34 | 0.04 $\pm$0.77 | **4.75** $\pm$0.45 |
| Eco-Cu$_2$ | *Lolium multiflorum* | 0.50 $\pm$0.69 | 0.36 $\pm$1.48 | 0.00 $\pm$6.00 | 0.72 $\pm$0.71 | 22.18 $\pm$0.59 | 0.02 $\pm$1.86 | 0.05 $\pm$0.72 | **2.02** $\pm$1.05 |
| Eco-Cu$_3$ | *Eichhornia crassipes* | 0.84 $\pm$0.57 | 0.26 $\pm$0.95 | 0.05 $\pm$1.82 | 0.14 $\pm$0.34 | 0.61 $\pm$6.97 | 0.81 $\pm$0.21 | 0.03 $\pm$5.86 | **10.37** $\pm$0.23 |
| Eco-Cu$_4$ | Functionalized coffee grounds | 0.46 $\pm$0.74 | 0.62 $\pm$0.63 | 0.06 $\pm$5.30 | 0.10 $\pm$1.04 | 8.81 $\pm$0.08 | 0.08 $\pm$2.31 | 0.17 $\pm$5.81 | **52.51** $\pm$2.16 |

It appeared that the functionalized coffee grounds could absorb by biosorption more copper element than the plants by rhizofiltration. This made coffee grounds a promising material for recycling copper in effluents. However, the lack of physiological metallic elements (for example Fe and K)

in it decreased the catalytic activity, in comparison to the Eco-Cu prepared from plants. Therefore, the Infrared (IR) study of the Lewis and Brønsted acidic character of Eco-Cu$_4$ was not carried out. The polymetallic composition of the other Eco-Cu was clearly an advantage.

### 3.2. Identification of the Degree of Oxidation

Usually, X-ray Photoelectron Spectrometry (XPS) allows the determination of the oxidation state of a metallic centre. In the case of copper salts, the XPS analysis did not allow to identify the oxidation's state, as the method induces the reduction of Cu(II) salts to Cu(I). The oxidation' states were established by different colourimetric methods, the oxime [73] and ammonia tests [74].

Both tests supported the presence of Cu(II) species in all catalysts.

### 3.3. Direct-Injection Mass Spectrometric Analysis of Eco-Cu$_3$

In order to specify the nature of the copper(II) chloride salts present in Eco-Cu$_3$, direct injection electrospray mass spectrometry operating in negative ion mode (MS-ESI) analyses were carried out. From Eco-Cu$_3$, two copper chloride species were detected: $CuCl_3^{2-}$ and $CuCl_2^{-}$. During the experiment, an electrochemical reduction of Cu(II) into Cu(I) was observed. This led to the conclusion that Eco-Cu$_3$ consisted of a mixture of $CuCl_4^{2-}$ and $CuCl_3^{-}$.

### 3.4. Morphology Study of Eco-Cu$_3$ by BET Analyses

BET analyses were performed in order to study the morphology of the new materials. The surface properties of the Eco-Cu$_3$ catalyst were determined by nitrogen sorption. The material has a relatively low porosity with an specific surface area $S_{BET}$ of 12.0 $m^2.g^{-1}$ and a mesoporous volume $V_{meso}$ of 0.07 $cm^3.g^{-1}$. An absorption of nitrogen appeared at high $P/P_0$ values, indicating a broad pore size distribution and an average pore size greater than 10 nm, due in part to intergranular porosity. Overall, bulk copper(I) catalyst exhibit a small surface area (for example 1.9 $m^2.g^{-1}$ with CuO). However copper nanoparticles with a defined size can have a greater surface area (136 $m^2.g^{-1}$), which allows for enhanced catalytic activity in an Ullmann reaction [9].

### 3.5. XRD Analysis of the Eco-Cu$_3$ Catalyst

X-ray Diffraction (XRD) analyses were performed in order to determine the crystalline structure of the different complexes in the Eco-Cu$_3$ catalyst. No crystalline form of copper was observed. However, a variety of salts such as $K_6Fe_2O_5$, MnO, NaCl, and $CaSO_4$ were detected. Among them, $K_6Fe_2O_5$, a polymetallic species, was reported by Shanks et al. as an efficient catalyst for the ethylbenzene dehydrogenation into styrene [75].

### 3.6. Analysis of the Acidic Properties of the Eco-Cu

#### 3.6.1. Lewis and Brønsted Acidic Character

The Lewis and Brønsted acid properties of the ecocatalyst Eco-Cu$_3$ were examined and compared to commercially available copper chloride. A first method of acidic properties analysis was based on the IR study of pyridine sorption/desorption at 25 °C and then 150 °C in order to distinguish the physisorbed pyridine from pyridine coordinately bonded to Lewis acid sites [76,77]. Pyridine is commonly used as a probe to evaluate the acidity of known Lewis and Brønsted acids by controlling its infrared absorption bands observed between 1400 and 1660 $cm^{-1}$. Absorption bands between 1445–1460 $cm^{-1}$ are characteristic for strongly bound Lewis acid sites of pyridine. The frequencies of these absorption bands were similar in the four catalysts (Table 4). Only in the case of Eco-Cu$_2$ the values were lower, suggesting a weaker Lewis acid character for this ecocatalyst. It was concluded that the Lewis acid character of Eco-Cu$_{1,3}$, anhydrous $CuCl_2$, and $CuCl_2 \cdot 2H_2O$ was similar. Surprisingly, in the case of Eco-Cu$_1$ and Eco-Cu$_3$, we have also observed absorption bands at 1529 and 1530 $cm^{-1}$.

This observation was interpreted as a proof for the unique Brønsted acid character of that ecocatalyst, not observed either in the case on commercially available copper chloride.

**Table 4.** The IR spectra of adsorbed pyridine on commercial anhydrous $CuCl_2$, on $CuCl_2 \cdot 2H_2O$, and on Eco-$Cu_{1-3}$.

| Catalyst | Lewis Acidity (1445–1460 cm$^{-1}$) | Lewis Acidity (1600–1640 cm$^{-1}$) | Brønsted Acidity (1500–1540 cm$^{-1}$) |
|---|---|---|---|
| $CuCl_2$ | 1450 | 1606 | - |
| $CuCl_2 \cdot 2H_2O$ | 1449 | 1605, 1635 | - |
| Eco-$Cu_1$ | 1449 | 1602, 1633 | 1529 |
| Eco-$Cu_2$ | 1447 | 1607 | - |
| Eco-$Cu_3$ | 1449 | 1606, 1645 | 1530 |

### 3.6.2. Analysis of the Lewis and Brønsted Acid Properties by the Corma Method

The results and information obtained during IR analysis were supported by the results of Corma's test performed in the related work. This method introduced by Corma et al. consists of studying the rearrangement pathway of the cyclic α-bromopropiophenone acetal in the presence of a catalyst [78]. The proportion of the rearrangement's products provides useful information evaluating the hardness of the Lewis acid character of the catalyst (Table 5). The formation of an ester results from the opening of the acetal by a hard Lewis acid. In contrast, the formation of the cyclic product proceeds from the action of a soft Lewis acid. Moreover, the Brønsted acidity of the catalyst can be evaluated by measuring the amount of propiophenone formed. The results using Corma's method for Eco-$Cu_3$ and $CuCl_2$ catalysts are presented below.

**Table 5.** The analysis of the acidic properties of the copper catalysts by Corma's method.

| Catalysts | Conversion Rate (%) [1] | Brønsted Acidity Products (%) [1] | Hard Lewis Acidity Products (%) [1] | Soft Lewis Acidity Products (%) [1] |
|---|---|---|---|---|
| Anhydrous $CuCl_2$ | 49 | 52 | 41 | 7 |
| $CuCl_2.2H_2O$ | 65 | 39 | 58 | 3 |
| Eco-$Cu_3$ | 100 | 64 | 36 | 0 |

[1] Determined by GC-MS analysis.

The first remarkable observation was the high reactivity of Eco-$Cu_3$ compared to both anhydrous and hydrated $CuCl_2$. In the presence of Eco-$Cu_3$, the complete conversion of the starting material was observed, while the use of both cupric chlorides led to a lower conversion, 49% and 65% depending on their degree of hydration. It appeared that the Brønsted acidic character was slightly higher for Eco-$Cu_3$ than in the case of the two commercially available copper chlorides, which supports the results obtained by IR. Finally, the formation of the heterocycle resulting from a soft acidity was observed only in the case of commercial copper chlorides. From this study, it was concluded that the Eco-$Cu_3$ was a stronger and harder Lewis acid than the commercially available cupric chlorides. This reactivity was rationalised with the polymetallic composition of the ecocatalysts and with the presence of other elements known to be hard Lewis acids, such as iron or calcium species, as shown by ICP-MS analysis.

## 4. Study of the Synthetic Potential of the Eco-Cu Catalysts

### 4.1. Cu-Catalysed Hydrolysis of Thiophosphates

The copper-catalysed hydrolysis of thiophosphates using an Eco-Cu has been reported using an example of environmental interest, namely the catalysed decomposition of parathion [68]. Discovered by Schrader, this compound has been widely used as an insecticide and acaricide agent [79]. However, parathion was recently found to be highly neurotoxic for many other organisms, as a good inhibitor of the enzyme acetylcholinesterase [80,81]. The transformation of parathion into paraoxon in the liver or in the environment is even more dangerous [82]. Therefore, their first identified objective was the development of a method to decompose such biocides (parathion here) in order to decontaminate humid or aquatic zones while being environmentally friendly technologies. A first difficulty was related to the chemical nature of this type of biocide. Indeed, the presence of the sulphur atom of the thiophosphate group is responsible for the much higher stability compared to a phosphoric ester [83]. Therefore, the proposed strategy for the rapid decomposition of parathion in aqueous media was based on the use the natural affinity of sulphur to coordinate with the Cu(II) ions. Copper should act as an electrophilic activator, weakening the P=S bond and facilitating the nucleophilic attack of water (Scheme 1). To experimentally confirm the utility of the Losfled strategy, the ecocatalyst (Eco-Cu$_1$) has been used as the source of copper. The Eco-Cu$_1$ was prepared from the biomass of a metallophyte plant, *Bacopa monierri*, able to bioaccumulate cooper during rhizofiltration of industrial effluents polluted with copper [69]. During the hydrolysis of parathion, the pH is usually important, especially to follow the formation of p-nitrophenolate. In the presented case, the pH during hydrolysis was close to 6 and could not promote the hydrolysis.

**Scheme 1.** The copper-catalysed hydrolysis of parathion.

Therefore, this developed process proposed the remediation of two kinds of pollution: (i) metallic pollution by Cu(II) salts where rhizofiltration can naturally clean contaminated water and (ii) pollution with organophosphate. This method is bio-based: The Eco-Cu$_1$ are prepared from *Bacopa monierri* and copper, the latter being overconcentrated because of anthropogenic metallurgy. The ecocatalysts are then used as a catalyst in the hydrolysis, and thus in the neutralization, of the toxic biocide parathion. A kinetic study of the hydrolysis of parathion was performed to evaluate the catalytic effect of these new plant-based copper species, according to the mechanism indicated above (Scheme 1). The progress of copper-catalysed hydrolysis of parathion was monitored by phosphorus nuclear magnetic resonance spectroscopy ($^{31}$P NMR) in order to unambiguously distinguish and quantify the numerous phosphorylated compounds present in the reaction mixture. Each phosphorylated compound was identified according to its chemical shift on the $^{31}$P NMR spectra which was strongly dependent from the electronic environment around the phosphorus atom (Figure 2).

The analysis of the $^{31}$P NMR spectra was demonstrative after 30 hours of stirring: the catalyst promoted the hydrolysis of parathion (29% remained instead of 56%, based on the integration of the signals on the $^{31}$P NMR) and the formation of diethyl thiophosphate. This decomposition was unambiguous, and no trace of paraoxon was observed (Figure 2).

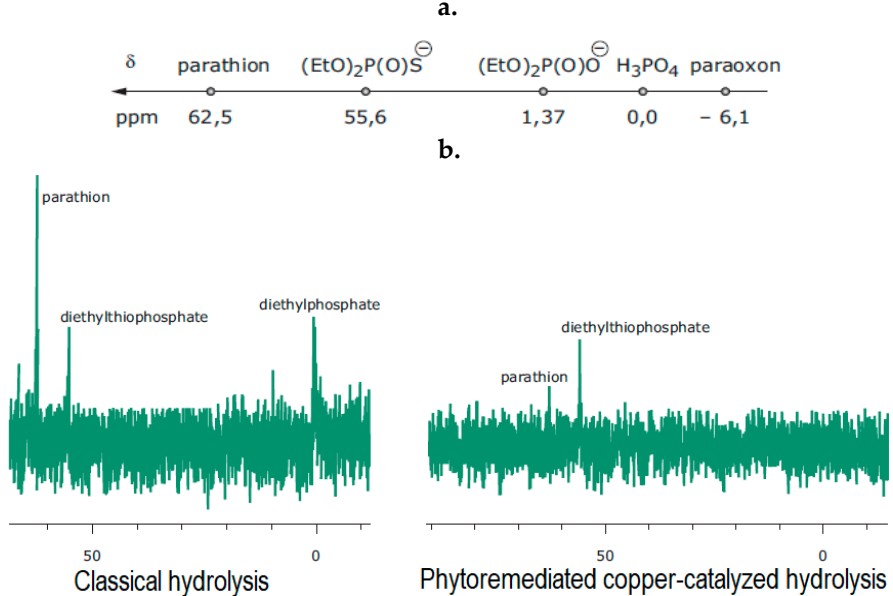

**Figure 2.** (**a**) The $^{31}$P NMR chemical shift of the phosphorylated species and (**b**) the $^{31}$P NMR study of the hydrolysis of parathion with commercially available $Cu^{2+}$ species (on the left) and the Eco-Cu prepared from the plant *Bacopa monnieri* after the rhizofiltration of copper(II) (on the right).

### 4.2. Copper-Catalysed Ullmann Coupling Reactions

As mentioned in the introduction, copper-catalysed coupling reactions are widely used in organic syntheses. The application of the Eco-Cu catalysts was investigated for Ullmann-type reactions [69]. Despite considerable efforts studying the Ullmann reaction, the development of ligands that are more efficient, more air and moisture stable, recyclable, and easy to prepare is still under investigation in order to facilitate this coupling reaction under green conditions. Copper-catalysed Ullmann coupling frequently requires harsh conditions, specific ligands, and large amounts of Cu for the reaction to occur. In this context, the catalytic potential of Eco-Cu was, therefore, particularly appealing.

#### 4.2.1. *N*-Arylation Ullmann-type Reaction

One of the particularities of Eco-Cu is their polymetallic structure. Those catalysts were, therefore, compared to the catalysts described by Taillefer et al. who have developed one of the few examples of bimetallic catalysis [84,85]. In the following described work, the *N*-arylation reaction carried out with pyrazole and iodobenzene was chosen as the comparative model (Table 6) [69]. Each Eco-Cu was found to be efficient for this reaction. A direct correlation between the coupling efficiency and the copper concentration in the catalyst was established. Importantly, at equal amounts of copper, Eco-Cu were found to be more active than $CuCl_2$. Only 1 mol. % of copper was enough to observe a good reactivity, which is a very low catalytic loading compared with reported studies on copper-catalysed coupling (10 to 20 mol. %). These results are described as an important advantage from the practical, environmental, and economic point of view. Additionally, Eco-Cu$_3$, which contains more copper than the other ecocatalysts, was found to be also the most active of all tested Eco-Cu.

Similar reactions under ligand-free conditions, reported in the literature, required a considerably higher loading of copper and harsher conditions, for example, 20 mol. % of Cu, 8 h at 120 °C in propionitrile as described by Hu et al. [86] or 10 mol. % of Cu, 24 h at 120 °C in DMF as described by Zhang et al. [87]. In comparison, the same authors reported the *N*-arylation of pyrazole with iodobenzene using excess tetraethylenepentamine (TEPA) (2 equiv.) as a base, and 10 mol. % of the copper catalyst was still needed or tetrabutylammonium bromide (TBAB) (3 equiv.) and a reaction time of 12 h at 125 °C was needed [88]. The copper loading could be reduced to 5 mol. % in some cases using unusual copper sources such as CuO hollow nanospheres immobilized on acetylene black [89] or

Cu nanoparticles [90]. However, both protocols required harsh conditions: 18 h at 180 °C and 18 h at 150 °C. Finally, the loading of 0.08 mol. % of copper for the *N*-arylation of pyrazole with iodobenzene under ligand-like conditions was reported by Bolm et al. [91]. However, this method required the presence of 1,2-dimethylethylenediamine (DMEDA) (20 mol. %); plus, the reactions were carried out at 135 °C for 24 h. It appeared that the use of Eco-Cu catalysts represented an improvement over the methods found in the literature. Once again, the performance of the ecocatalysts was explained by their polymetallic composition and, more precisely, by the presence of alkaline metals ($Na^+$ and $K^+$) and salts with Lewis acid properties ($Ca^{2+}$, $Mg^{2+}$, and $Fe^{3+}$). This hypothesis was in agreement with the work of Zhang et al. who demonstrated the advantages of inorganic salt particles in transition metal-catalysed coupling reactions [92]. Partial negative charges on the salt surface created an electron donor effect and increased the electron density around the metal centres. In addition, the anions on the surface of the material have a similar influence on the aryl halide. The combination of these effects favoured the oxidative addition step and improved the reaction rate. In addition, according to Fan et al., the presence of Lewis acids would favour the polarization of the aryl-halogen bond [93]. This effect resulted in a significant increase of the cross-coupling rate [94]. Finally, these results were consistent with the previous work reported by Grison and coworkers on Heck-Mizoroki and Suzuki-Miyaura cross-coupling reactions with Eco-Pd catalysts, characterized by an excellent dispersion of the active centres on the mineral matrix of Na, K, Ca, Mg, and Fe [95].

**Table 6.** A comparison of the catalytic activity of Eco-Cu and $CuCl_2$.

| Entry [1] | [Cu] Catalyst | wt. % of Eco-Cu | [Cu] Quantity (mol. %) | Yields [2] |
|---|---|---|---|---|
| 1 | Eco-Cu$_1$ | 4.8 | 1 | 84 |
| 2 | Eco-Cu$_2$ | 2.0 | 1 | 77 |
| 3 | Eco-Cu$_3$ | 10 | 1 | 85 |
| 4 | Eco-Cu$_3$ | 10 | 0.25 | 57 |
| 5 | CuCl$_2$ | - | 1 | 48 [3] |
| 6 | CuCl$_2$ | - | 3 | 79 [3] |

[1] Reaction conditions: iodobenzene (5 mmol, 1 eq.), pyrazole (7.5 mmol, 1.5 eq.), Eco-Cu (x mol. %), $Cs_2CO_3$ (10 mmol, 2 eq.), DMF (5 mL), 90 °C, 15 h, and under argon atmosphere. [2] Isolated yields. [3] Yields were determined by GC-MS analysis.

This Ullmann-type reaction was extended to other azole derivatives but with some limitations concerning aniline and secondary amines such as pyrrolidine and morpholine as nucleophiles (Table 7).

**Table 7.** The extension of the Ullmann reaction to other nitrogen nucleophiles.

| Entry | Nitrogen Nucleophile | Yields (%) [1,2] |
|---|---|---|
| 1 | Pyrazole | 85 |
| 2 | Imidazole | 54 |
| 3 | 2-Pyrrolidinone | 31 [3] |
| 4 | Aniline | 0 [3] |
| 5 | Pyrrolidine | 6 [3] |
| 6 | Morpholine | 5 [3] |

[1] Reaction conditions: iodobenzene (5 mmol, 1 eq.), pyrazole (7.5 mmol, 1.5 eq.), Eco-Cu$_3$ (1 mol. %), $Cs_2CO_3$ (10 mmol, 2 eq.), DMF (5 mL), 90 °C, 15 h, and under argon atmosphere. [2] Isolated yields. [3] The yields were determined by GC-MS analysis.

The study of the scope of the reaction showed a good applicability of the method with a large variety of aryl halides, substituted by electron withdrawing or donating groups (Table 8). The use of DMF as a solvent was required; only $\gamma$-valerolactone could replace DMF, but lower conversions were observed. The Eco-Cu$_3$ promoted efficient coupling reactions with aryl iodide, bromide, and even chloride. The yields were good to excellent when the temperature was correctly adjusted. As expected, the presence of an electron donating group such as p-OMe resulted in a decrease of reactivity (entry 2). However, by increasing the temperature of the reaction from 90 to 110 °C (entry 3), the aryl halide, even substituted with an electron donating group, could react efficiently with the pyrazole partner. Finally, the presence of electron-withdrawing groups facilitated the coupling reaction (entries 4–9).

**Table 8.** The extension of the Ullmann reaction to other halogen derivatives.

R = OMe, Ac, NO$_2$, CN
X = Cl, Br, I

| Entry | Aryl Halide | T (°C) | Time (h) | Yield (%) [1,2] |
|-------|-------------|--------|----------|-----------------|
| 1 | R = H; X = I | 90 | 15 | 85 |
| 2 | R = OMe; X = I | 90 | 15 | 63 |
| 3 | R = OMe; X = I | 110 | 15 | 93 |
| 4 | R = COMe; X = I | 90 | 4 | >98 |
| 5 | R = COMe; X = Br | 90 | 15 | 31 |
| 6 | R = NO$_2$; X = Br | 90 | 4 | >98 |
| 7 | R = NO$_2$; X = Cl | 90 | 4 | >98 |
| 8 | R = CN; X = Cl | 90 | 15 | 73 |
| 9 | R = CN; X = Cl | 110 | 15 | 89 |

[1] Reaction conditions: aryl halide (5 mmol, 1 eq.), pyrazole (7.5 mmol, 1.5 eq.), Eco-Cu$_3$ (1 mol. %), Cs$_2$CO$_3$ (10 mmol, 2 eq.), DMF (5 mL), T °C, time (h), and under argon atmosphere. [2] Isolated yields.

### 4.2.2. O-Arylation Ullmann-type Reaction

The creation of C–O bonds by Ullmann reaction is less reported in the literature than the *N*-arylation. The most conventional approach is based on the reaction of aryl bromides with phenols in the presence of ligands and a high catalyst loading (10–30 mol. %) and at elevated temperatures (50–110 °C). In the case of the use of aryl chlorides, even higher temperatures are necessary (135 °C to 160 °C). Therefore, the study of the potential application of Eco-Cu catalysts for that reaction was interesting (Table 9).

The *O*-arylation reactions catalysed by Eco-Cu$_3$ were found to be very efficient in the presence of Cs$_2$CO$_3$ as a base at 110–130 °C. The reaction did not require the presence of any ligand, and only 1 mol. % of copper was sufficient to reach very good yields (51–98%) and a large substrate scope. The high potential of Eco-Cu to catalyse efficiently Ullmann-type coupling reactions has been demonstrated. Importantly *N*- and *O*-arylation reactions did not work without the Eco-Cu catalyst or with CuO [96].

**Table 9.** The Eco-Cu$_3$-catalysed *O*-arylation.

| Entry | Aryl Halide | T (°C) | Product | Yield (%) [1,2] |
|:---:|:---:|:---:|:---:|:---:|
| 1 | R$^1$ = H; X = I | 110 | | 66 [3] |
| 2 | R$^1$ = H; X = I | 110 | | 64 [3] |
| 3 | R$^1$ = RCOMe; X = I | 110 | | >98 |
| 4 | R$^1$ = H; X = I | 130 | | >98 |
| 5 | R$^1$ = H; X = I | 130 | | >98 |
| 6 | R$^1$ = H; X = Br | 130 | | 84 |
| 7 | R$^1$ = H; X = Br | 130 | | 82 |
| 8 | R$^1$ = H; X = Br | 130 | | 92 |
| 9 | R$^1$ = H; X = Br | 130 | | 51 [3] |
| 10 | R$^1$ = NO$_2$; X = Br | 110 | | >98 |
| 11 | R$^1$ = NO$_2$; X = Cl | 110 | | >98 |
| 12 | R$^1$ = NO$_2$; X = Cl | 110 | | >98 |

[1] Reaction conditions: Aryl halide (5 mmol, 1 eq.), phenol (6 mmol, 1.2 eq.), Eco-Cu$_3$ (1 mol. %), Cs$_2$CO$_3$ (10 mmol, 2 eq.), DMF (5 mL), 15 h, and under argon atmosphere. [2] Isolated yields. [3] The yields were determined by GC-MS analysis.

### 4.3. The Copper(I)-Catalysed Alkyne-Azide Cycloaddition (CuAAC) "Click" Reaction

As a last huge application of copper catalysis, the feasibility with Eco-Cu of 1,3-dipolar cycloaddition between azides and alkynes was investigated [70]. This important reaction was the first example of "click chemistry" where the mechanism of "fusion" between two molecules represents a perfect atom economy [97]. This reaction fits perfectly into the current trends in green chemistry. It is also a very good method of access to substituted 1,2,3-triazoles. Finally, CuAAC has led to many applications in the fields of organic synthesis [98], medicinal chemistry [99,100], molecular biology [99,101], and materials science [102]. The "click" chemistry is increasingly used for the labelling of biomacromolecules [39,103,104] or as functional tool to understand complex biological systems [105,106]. The abovementioned facts illustrate the current interest for synthesizing triazoles using aqueous systems. Additionally, in the context of green chemistry, the development of ligand-free catalysts is an additional challenge. Ligand free catalysis represents a double advantage in terms of cost (the high cost of the ligands limits the industrial interest of the process) and purification (the delicate separation step between the ligand and the product is avoided). The Eco-Cu represented a very good alternative for the development of new sustainable and environmentally friendly catalytic processes. This possibility was demonstrated by of Eco-Cu$_3$ catalyzed azide-alkyne cycloaddition (CuAAC) in aqueous medium [70]. Additionally, for the first time, it has been shown that a homogeneous Cu-based catalyst, Eco-Cu$_3$, can be recycled by the rhizofiltration technique from the reaction mixture and then reused.

### 4.3.1. Application of Eco-Cu in the CuAAC Reaction

The reaction of benzyl azide with propargyl alcohol was chosen as the illustrative example. During the optimisation of the reaction, it appeared that sodium ascorbate, a natural and aqueous soluble reducing agent, could reduce in situ Eco-Cu$_3$ into a Cu(I) active catalyst in only few minutes at room temperature. The Eco-Cu$_3$ was found to be very efficient for this reaction, using a mixture of triethyl amine and sodium ascorbate (NaAsc). Different solvent systems were investigated (Table 10): Very good conversions were obtained in green solvents, such as water, ethanol, isopropanol, or 2-methyltetrahydrofuran.

**Table 10.** The influence of the solvent on the CuAAC Click reaction with Eco-Cu$_3$.

| Entry | Solvent | Conversion (%) [1,2] |
|---|---|---|
| 1 | $H_2O$ | >99 |
| 2 | EtOH | >99 |
| 3 | *i*-PrOH | >99 |
| 4 | 2-Me-THF | >99 |
| 5 | DMF | >99 |
| 6 | $H_2O$ [3] | 0 |

[1] Reaction conditions: Propargyl alcohol (1 mmol, 1 eq.), azide (1 mmol, 1 eq.), Eco-Cu$_3$ (5 %mol.), NaAsc (5 mmol, 5 eq.), Et$_3$N (20 mmol, 20 eq.), solvent, 25 °C, and 16 h. [2] The conversions were determined by GC-MS after calibration with hexadecane as an internal standard. [3] A blind experiment in which the ecocatalyst has been prepared following the same protocol from a plant that was never in contact with the CuNO$_3$ solution.

The polymetallic composition of Eco-Cu$_3$ did not affect the interactions between the copper site and the substrate. Additionally, the ecocatalyst displayed a very high selectivity, and no formation of by-products was observed. The reaction led exclusively to 1,4-substituted triazole. It has to be noted that no Glaser coupling product (diynes) was detected under aerobic conditions. However, in the presence of K$_2$CO$_3$ under air, the only observed reaction is the Glaser coupling product, 2,4-hexadiyne-1,6-diol, resulting from the homocoupling of the alkyne. The most efficient conditions

reported in the literature for CuAAC involved the use of $CuSO_4$ under aqueous conditions or CuI, CuBr, $Cu(OAc)_2$, or Cu in organic solvents [107]. The effectiveness of CuCl or $CuCl_2$/sodium ascorbate was rarely described [102]. The use of Eco-Cu$_3$ ensured very simple experimental conditions and a large application of the method to aliphatic, aromatic, and functionalised substrates such as propargyl ester, alkenyl, or substrates containing amines or alcohol groups (Table 11). Worth mentioning is the activity of the Eco-Cu$_3$, comparable to homogeneous Cu catalysts [108], and recently to the heterogeneous catalysts described by Taskin et al. [109]. The clear advantage of this methodology reported here with Eco-Cu was the simplicity of the reaction treatment and isolation of products. No purification was necessary: A simple addition of 2-methyltetrahydrofurane or ethyl acetate allowed the isolation of the products with a high purity.

**Table 11.** A substrate scope of the Eco-Cu-catalysed CuAAC reaction.

| Entry | Azide | Alkyne | Yield (%) [1,2] (with Eco-Cu$_3$) | Yield (%) (other [Cu]) |
|---|---|---|---|---|
| 1 | Me~~~N$_3$ | Ph | >99 (92)[3] | >99 [110] |
| 2 | t-Bu–O–C(O)–O–N$_3$ | n-Hex | >99 | - |
| 3 | Ph~N$_3$ | Ph | >99 (78)[3] | >99 [110] |
| 4 | Ph~N$_3$ | (propargyl acrylate) | 98 | 98 [109] |
| 5 | Ph~N$_3$ | OH | >99 | 61 [109] |
| 6 | EtO–C(O)–CH(Me)–N$_3$ | Ph | >99 | 85 [109] |
| 7 | (anthracenylmethyl)–N$_3$ | Ph | >99 | 93 [109] |
| 8 | Ph~N$_3$ | NH$_2$ | >99 | 79 [110] |
| 9 | Ph~N$_3$ | COOH | >99 | >99 [111] |
| 10 | EtO–C(O)–CH(Me)–N$_3$ | n-Hex | >99 | - |

[1] Reaction conditions: Propargyl alcohol (1 mmol, 1 eq.), azide (1 mmol, 1 eq.), Eco-Cu$_3$ (5 mol. %), NaAsc (2 mmol, 2 eq.), H$_2$O, 25 °C, and 16 h. [2] The conversions were determined by GC-MS after calibration with hexadecane as an internal standard. [3] Isolated yield.

### 4.3.2. Recycling and Reuse of the Ecocatalysts

An ecotechnology (rhizofiltration or biosorption) was performed with the plant *E. crassipes* of the waste aqueous phase after treatment of the CuAAC reaction reported above. It has been reported that the plant could bioaccumulate the metals from the first ecocatalyst used. During such processes, 95% and 53% of the Cu were bioaccumulated, respectively (rhizofiltration of biosorption). After the usual thermal and acidic treatment, the recycled ecocatalysts Eco-Cu$_4$ and Eco-Cu$_5$ were obtained and characterised (Figure 3). The ICP-MS analysis revealed a significant level of iron (1.35 wt. %), that represented more than a half of the amount of copper (2.29 wt. %). An X-ray analysis revealed that the

Eco-Cu$_5$ structure was very close to the one of Eco-Cu$_3$. The catalytic activity of the recycled Eco-Cu$_5$ was tested in a CuAAC reaction using benzylazide and phenylacetylene as substrates. Under the optimised conditions (described above), the reaction remained quantitative. This result was noteworthy as it is one of the rare cases of the recycling and reuse of a homogeneous catalyst. The ecological conditions and the effectiveness of the methodology are, therefore, remarkable.

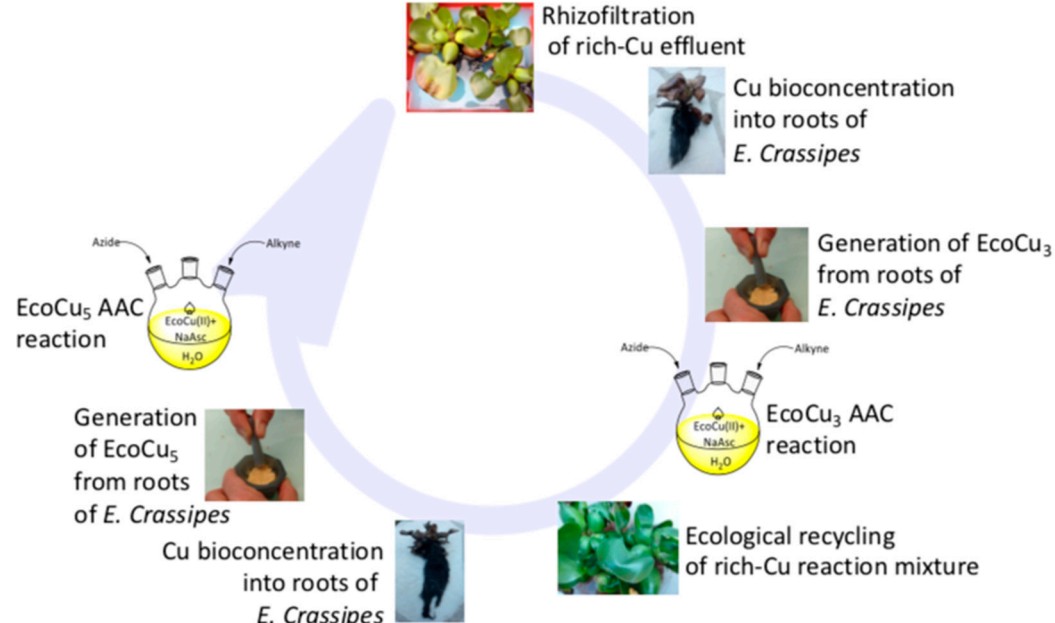

**Figure 3.** The recycling and reuse of the Eco-Cu$_3$.

## 5. Conclusions

The first examples of Cu catalysts of plant origin, Eco-Cu, were presented in this review. These catalysts resulted from the bioaccumulation of copper by living (rhizofiltration) or dead (biosorption) aquatic plants. Among the aquatic plants used, the *Eichhornia crassipes* was found to be the most efficient to such purposes and, in each case, allowed the innovative recovery of Cu from copper-rich effluents. The transformation of Cu-rich roots led to Eco-Cu catalysts. The polymetallic composition of the Eco-Cu was established as the reason for their interesting, original, and unique catalytic activity. The utility of the Eco-Cu was demonstrated through three main reactions for copper catalysis: (i) the catalytic hydrolysis of the thiophosphate group; (ii) the Ullmann-type reactions and their application for *N*- and *O*-arylation; and (iii) finally, the CuAAC "click" reaction. These reactions were very simple and did not require the presence of an amine or ligands. Importantly, the obtained products did not require any additional purification. A very important aspect of the use of Eco-Cu was the possibility of their recovery and reuse. The recovery of the catalyst could be effectively preformed via ecological recycling of the reaction mixture through a second cycle of bioaccumulation. The recycled Eco-Cu has a similar catalytic activity. This easy recovery of Eco-Cu is another asset in terms of green catalysis.

**Author Contributions:** Writing—review and editing, T.K.O., P.A., and C.G.

**Acknowledgments:** We thank the Université de Montpellier and the CNRS for financial support.

**Conflicts of Interest:** The authors declare no conflict of interest.

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
