# Peer review of "Bio-based Catalysts from Biomass Issued after Decontamination of Effluents Rich in Copper—An Innovative Approach towards Greener Copper-based Catalysis"

_catalysts, doi:10.3390/catal9030214_

Round 1
Reviewer 1 Report
This review deals with (i) the use of plants that can concentrate copper (amongst other metals) in polluted water, (ii) how to extract, obtain new copper catalysts and (iii) and the applications of those new eco-copper catalysts in different selected reactions.
The review is really well written and is of high interest for a broad range of researchers since discussion about decontamination of polluted water, characterization of catalyst and their use in known catalytic reactions. All the sections will be helpful to the scientific community.
I just have some minor remarks to further help the future readers :
- Page 2, line 70 : to help the reader, I’ll add a quick definition for rhizofiltration and biosorption.
- Page 3, line 87 - Could you precise why those specific plants in particular ? Does it exist other aquatic plants or non-aquatics that could work as well?
- Page 4, line 130. What I would like to read in a review is some discussion. The method to recover the catalysts is it really easy to handle and to reproduce? Have the authors tried to reproduce it? If yes, they should probably precise if there are any tricks to succeed in the extraction of the Eco-Cu.
- Page 5, line 162. Some discussion could be added for the BET analyses. Why the specific surface area have only been calculated for Eco-Cu3? The Sbet is quite small. A comment will be nice and a comparison with others catalysts described in literature would be interested.
- Page 5, line 183, there is a problem within this sentence.
- Page 5, Table 4, Why FTIR of Eco-Cu4 has not been recorded ?
- Page 6, line 194, “of the catalyst can be”. Be is missing
- Page 7, line 228 and 242, problem in the formatting
- Page 7, scheme 1, for the hydrolysis of parathion, pH is usually important, especially to follow the formation of p-Nitrophenolate. Could the authors precise the pH for this hydrolysis?
- Page 7, line 245, it is figure 2a and not 1a
I have a concern for this reaction. How to be sure that it is only the copper working for the hydrolysis? Has it been tried with other Eco-Cu ?
It is stated that Eco-Cu1 is use for this hydrolysis, but Eco-Cu4 is much richer in copper. Has it been tried?
- Page 9, line 284, precise what is TBAB
- Page 12, line 354, a space is needed between above and mentioned facts
- Page 12, table 10, a blank reaction with water rich in copper has been ever tested, with no isolation of the Eco-Cu ?
- Page 13, line 407, problem in the number for the recycled catalysts because Eco-Cu4 already exist?
- Page 14, line 409, a comment on the significant level of iron? Where does that come from?
- Page 14, problem in Figure 3
Author Response
Dear Editor,
Please find attached the new version of our previous manuscript, initially submitted under the manuscript number catalysts-439451 in Catalysts. We would like to thank you for giving us the possibility to resubmit our paper, in accordance with your mail of February 18, 2018. We also would like to thank the referees for their comments, which we have read with great care and interest. We believe that we have addressed the referee’s comments and have attached the necessary changes below. We feel these comments and the changes resulting from these have improved the quality of the manuscript.
Reviewers' Comments to Authors:
Reviewer 1:
Comments and Suggestions for Authors
This review deals with (i) the use of plants that can concentrate copper (amongst other metals) in polluted water, (ii) how to extract, obtain new copper catalysts and (iii) and the applications of those new eco-copper catalysts in different selected reactions.
The review is really well written and is of high interest for a broad range of researchers since discussion about decontamination of polluted water, characterization of catalyst and their use in known catalytic reactions. All the sections will be helpful to the scientific community.
I just have some minor remarks to further help the future readers:
- Page 2, line 70: to help the reader, I’ll add a quick definition for rhizofiltration and biosorption.
Answer: The following two definitions were included in the manuscript text. „Rhizofiltration refers to the approach in which aquatic plant roots are used to purify contaminated water through metabolically mediated process. Whereas biosorption can be defined as the ability of biological materials to accumulate pollutants from wastewater through physicochemical pathways based on different mechanisms including absorption, adsorption, ion exchange, surface complexation or precipitation”
- Page 3, line 87 - Could you precise why those specific plants in particular? Does it exist other aquatic plants or non-aquatics that could work as well?
Answer: When choosing plants two criteria were selected: i) the biomass should be insoluble in water, ii) the structure of biomass should be based on carbon-containing aromatic compounds and contain many carboxylate groups naturally present in the material of plant origin.
- Page 4, line 130. What I would like to read in a review is some discussion. The method to recover the catalysts is it really easy to handle and to reproduce? Have the authors tried to reproduce it? If yes, they should probably precise if there are any tricks to succeed in the extraction of the Eco-Cu.
Answer: Recovery and recycling is described at the end of the article in part 4.3.2.
- Page 5, line 162. Some discussion could be added for the BET analyses. Why the specific surface area has only been calculated for Eco-Cu3? The Sbet is quite small. A comment will be nice and a comparison with others catalysts described in literature would be interested.
Answer: The following sentence was added to the paragraph 3.4. concerning BET analysis.
Overall, bulk copper(I) catalyst exhibit a small surface area (for example 1.9 m 2 .g − 1 with CuO). However copper nanoparticles with defined size can have a greater surface area (136 m2.g-1), which allows for enhanced catalytic activity in Ullmann reaction[9].
- Page 5, line 183, there is a problem within this sentence.
Answer: Sentence: “Surprisingly in the case of Eco-Cu1 and Eco-Cu3 and 1530 cm-1” was replaced with: “Surprisingly in the case of Eco-Cu1 and Eco-Cu3 we have also observed absorption bands at 1529 and 1530 cm-1”
- Page 5, Table 4, Why FTIR of Eco-Cu4 has not been recorded?
Answer: The following explanation was introduced below table 4:
“However, the lack of physiological metallic elements (for example Fe, K) in it decreased the catalytic activity, in comparison to the Eco-Cu prepared from plants. Therefore the Infra Red (IR) study of their Lewis and Brønsted acidic character of Eco-Cu4 was not carried out. The polymetallic composition of the other Eco-Cu was clearly an advantage.”
- Page 6, line 194, “of the catalyst can be”. Be is missing
Answer: This was corrected in the text.
- Page 7, line 228 and 242, problem in the formatting
Answer: The sentence: “The original idea reported by Losfeld et al. was first to exploit the capacity of a metallophyte plant, such as Bacopa monierri, to bioaccumulate (rhizofiltration) copper salts from polluted areas and then to transform this copper‑rich plant into ecocatalysts (Eco‑Cu1) to finally use it for the hydrolysis of the biocide [55].” Was replaced with: ”To experimentally confirm the utility of Losfled strategy, ecocalyst (Eco-Cu1) has been used as the source of copper. The Eco-Cu1 was prepared from biomass of metallophyte plant Bacopa monierri able to bioaccumulate cooper during rhizofiltration of industrial effluents polluted with copper [64].”
The sentence: ”The progress of the reaction was monitored by 31P NMR in order to unambiguously distinguish and quantify the numerous phosphorylated compounds. Species are identified according to their chemical shift (Figure 1a). Indeed, the electronic environment around each phosphorus atom allowed easy assignment of each signals to the corresponding compound (Figure 1b). Was replaced with: “The progress of copper-catalysed hydrolysis of parathion was monitored by phosphorus nuclear magnetic resonance spectroscopy (31P NMR) in order to unambiguously distinguish and quantify the numerous phosphorylated compounds present in the reaction mixture. Each phosphorylated compound was identified according to its chemical shift on the 31P NMR spectra which was strongly dependent from the electronic environment around the phosphorus atom (Figure 2).”
- Page 7, scheme 1, for the hydrolysis of parathion, pH is usually important, especially to follow the formation of p-Nitrophenolate. Could the authors precise the pH for this hydrolysis?
Answer: We included the following sentence just above the scheme 1:
“During the hydrolysis of parathion the pH is usually important, especially to follow the formation of p-nitrophenolate. In the presented case the pH during hydrolysis was close to 6 and could not promote the hydrolysis.”
- Page 7, line 245, it is figure 2a and not 1a
Answer: This was corrected. Indeed, this should be Figure 2.
I have a concern for this reaction. How to be sure that it is only the copper working for the hydrolysis? Has it been tried with other Eco-Cu?
It is stated that Eco-Cu1 is use for this hydrolysis, but Eco-Cu4 is much richer in copper. Has it been tried?
Answer: The presented review refers to the results reported in reference 64 and there the authors did not describe the catalytic activity of another Eco-Cu for that reaction.
- Page 9, line 284, precise what is TBAB
Answer: This was corrected. We explained the TBAB is tetrabutylammonium bromide.
- Page 12, line 354, a space is needed between above and mentioned facts
Answer: This was corrected.
- Page 12, table 10, a blank reaction with water rich in copper has been ever tested, with no isolation of the Eco-Cu?
Answer: We thank the reviewer for this comment. However, the reaction does not work under such condition.
- Page 13, line 407, problem in the number for the recycled catalysts because Eco-Cu4 already exist?
Answer: This was corrected. The recycled catalyst was marked as Eco-Cu5.
- Page 14, line 409, a comment on the significant level of iron? Where does that come from?
Answer: Iron is a physiological element present in the plant.
- Page 14, problem in Figure 3
Answer: This was corrected. The recycled catalyst was marked as Eco-Cu5.
Many thanks for taking the time to review our changes; we hope these modifications are in accord with the reviewer's and editor’s expectations.
Yours sincerely,
Claude Grison and the authors
Reviewer 2 Report
The manuscript submitted by C GRISON et al. is a review dealing with the preparation, characterization and catalytic applications of a new set of bio-sourced copper-based materials, named Eco-Cu by the authors. The present version is very well structured and clear, even for a non-expert in the field. The advantages of the Eco-Cu catalysts developed by the authors, especially within the Green Chem, are well demonstrated throughout the manuscript.
Nevertheless the quality of the manuscript, I have several suggestions/comments/questions in order to improve the present version:
INTRODUCTION
The case of copper-based heterogeneous – and thus recyclable – catalysts is surprisingly not mentioned in the introduction. A paragraph dealing with this aspect would be welcome here, through a brief mention of the main supports that have been successfully applied to the copper-catalyzed transformations considered here (ie, organic polymers & biopolymers, carbon materials such as charcoal & graphene, porous materials such as mesoporous silica, microporous zeolites & MOFs, …).
PARTS 4.2/4.3
ULLMANN-type coupling reactions (N-, O- & C-arylation) are known to be catalyzed by Cu(I) species. Do the Eco-Cu materials contain such Cu(I) species? If not, how do the authors explain the catalytic activity of these materials in these relevant coupling reactions? In contrast, sodium ascorbate was required as reducing agent in order to promote the CuAAC with these catalysts.
In the particular case of ULLMANN-type arylation reactions employing electron-poor aryl halides (Table 8, entries 4-9 – Table 9, entries 3, 10-12), did the authors perform control experiments in the absence of the catalyst? These examples could be due to pure SNAr reactions and such control experiments would be helpful in order to demonstrate that the reported reactions are unambiguously due to the presence of the catalyst.
In addition, in the diaryl ether series, the authors obtained excellent results using electron-poor phenols, in particular 4-cyanophenol (Table 9, entries 10 & 12). This result is remarkable because in most cases, electron-poor phenols are poor coupling partners for such reactions. However, 4-cyanophenol has only be coupled to electron-poor 4-nitrophenyl bromide/chloride. An additional example using phenyl iodide/bromide/chloride (and/or 4-methoxyphenyl iodide/bromide/chloride) as aryl halide would be relevant here. Similarly, 4-nitrophenol would also be interesting to be tested as electron-poor phenolic partner.
TYPOS
- Page 4 lines 137-138: … more copper element than the plants …
- Page 5 line 183: the sentence ‘Surprisingly in the … 1530 cm-1.’ is suspect
- Page 6 lines 194-195: … of the catalyst can be evaluated by measuring …
- Page 6 lines 225-226: the sentence ‘Therefore, … for Cu(II) ions.’ is suspect
Author Response
Dear Editor,
Please find attached the new version of our previous manuscript, initially submitted under the manuscript number catalysts-439451 in Catalysts. We would like to thank you for giving us the possibility to resubmit our paper, in accordance with your mail of February 18, 2018. We also would like to thank the referees for their comments, which we have read with great care and interest. We believe that we have addressed the referee’s comments and have attached the necessary changes below. We feel these comments and the changes resulting from these have improved the quality of the manuscript.
Reviewers' Comments to Authors:
Reviewer 2:
Comments and Suggestions for Authors
The manuscript submitted by C GRISON et al. is a review dealing with the preparation, characterization and catalytic applications of a new set of bio-sourced copper-based materials, named Eco-Cu by the authors. The present version is very well structured and clear, even for a non-expert in the field. The advantages of the Eco-Cu catalysts developed by the authors, especially within the Green Chem, are well demonstrated throughout the manuscript.
Nevertheless, the quality of the manuscript, I have several suggestions/comments/questions in order to improve the present version:
INTRODUCTION
The case of copper-based heterogeneous – and thus recyclable – catalysts is surprisingly not mentioned in the introduction. A paragraph dealing with this aspect would be welcome here, through a brief mention of the main supports that have been successfully applied to the copper-catalyzed transformations considered here (ie, organic polymers & biopolymers, carbon materials such as charcoal & graphene, porous materials such as mesoporous silica, microporous zeolites & MOFs, …).
Answer: This deficiency was corrected by adding to the introduction the following text along with suitable literature references. Furthermore, in line with the current trends in sustainable and green chemistry heterogenous catalysts for copper-based reactions were reported in the literature. This area of research continues to evolve as a more suitable approach because of its crucial advantages such as easy separation of products from catalyst, high stability of the heterogenous catalysts and, most importantly, their recyclability. In that aspect, the selected examples heterogenous copper catalysts employing Cu/C (copper-in-charcoal),[44,45] copper powder,[46] magnetite-supported copper nanoparticles, [47] Cu/ligand catalyst immobilized on silica,[48] CuI immobilized on MOF,[49] alumina-supported CuO,[50] and recent mesoporous copper supported manganese oxide material (meso Cu/MnOx)[51] as well as copper oxide catalysts supported on three dimensional mesoporous aluminosilicates[52] are certainly worth mentioning.
44. Lipshutz, B. H.; Unger, J. B.; Taft, B. R. Copper-in-charcoal (Cu/C) promoted diaryl ether formation. Org. Lett. 2007, 9, 1089− 1092.
45. Benjamin R. Buckley, Rachel Butterworth, Sandra E. Dann, Harry Heaney, and Emma C. Stubbs “Copper-in-Charcoal” Revisited: Delineating the Nature of the Copper Species and Its Role in Catalysis ACS Catal., 2015, 5, pp 793–796
46. Jiao, J.; Zhang, X.-R.; Chang, N.-H.; Wang, J.; Wei, J.-F.; Shi, X.- Y.; Chen, Z.-G. A facile and practical copper powder-catalyzed, organic solvent-and ligand-free Ullmann amination of aryl halides. J. Org. Chem. 2011, 76, 1180−1183.
47. Shelke, S. N.; Bankar, S. R.; Mhaske, G. R.; Kadam, S. S.; Murade, D. K.; Bhorkade, S. B.; Rathi, A. K.; Bundaleski, N.; Teodoro, O. M.; Zboril, R. Iron oxide-supported copper oxide nanoparticles (Nanocat-Fe-CuO): magnetically recyclable catalysts for the synthesis of pyrazole derivatives, 4-methoxyaniline, and Ullmann-type condensation reactions. ACS Sustainable Chem. Eng. 2014, 2, 1699−1706.
48. Benyahya, S.; Monnier, F.; Taillefer, M.; Man, M. W. C.; Bied, C.; Ouazzani, F. Efficient and Versatile Sol-Gel Immobilized Copper Catalyst for Ullmann Arylation of Phenols. Adv. Synth. Catal. 2008, 350, 2205− 2208.
49. Wang, M.; Yuan, B.; Ma, T.; Jiang, H.; Li, Y. Ligand-free coupling of phenols and alcohols with aryl halides by a recyclable heterogeneous copper catalyst. RSC Adv. 2012, 2, 5528−5530.
50. Ling, P.; Li, D.; Wang, X. Supported CuO/γ-Al2O3 as heterogeneous catalyst for synthesis of diaryl ether under ligand-free conditions. J. Mol. Catal. A: Chem. 2012, 357, 112−116.
51. Kankana Mullick, Sourav Biswas, Chiho Kim, Ramamurthy Ramprasad,Alfredo M. Angeles-Boza, and Steven L. Suib Ullmann Reaction Catalyzed by Heterogeneous Mesoporous Copper/ Manganese Oxide: A Kinetic and Mechanistic Analysis Inorg. Chem. 2017, 56, 10290−10297
52. Udayakumar Veerabagu, Gowsika Jaikumar, Pandurangan Arumugam, Sabarathinam Shanmugan, Lu Fushen An efficient copper catalyzed 3D mesoporous aluminosilicate for the synthesis of dibenzodiazonines in the Ullmann cross-coupling reaction New J. Chem., 2018, 42, 13065-13073.
PARTS 4.2/4.3
ULLMANN-type coupling reactions (N-, O- & C-arylation) are known to be catalyzed by Cu(I) species. Do the Eco-Cu materials contain such Cu(I) species? If not, how do the authors explain the catalytic activity of these materials in these relevant coupling reactions? In contrast, sodium ascorbate was required as reducing agent in order to promote the CuAAC with these catalysts.
Answer:
Thermal treatment of plants’ roots used to produce EcoCu determines the oxidation state of copper present in EcoCu: under air flow, Cu(II) species are exclusively formed.
In the case of CuAAC reaction, it is generally accepted that Cu(II) salts are not efficient catalysts in the CuAAC reaction. Sodium ascorbate, a natural and aqueous soluble reducing agent, is the reagent of choice to generate an active Cu(I) catalyst in situ. The treatment of yellow EcoCu (II) by sodium ascorbate yielded the gray-green EcoCu (I) in five minutes at room temperature.
XPS analysis was performed on Eco-Cu to study the oxidation state of copper after thermal treatment and activation with HCl and we could clearly see the expected presence of Cu(II). The presence of Cu(I) was also observed. However, we assume that the observation of Cu(I) is not due to Eco-Cu, because it is well known that Cu(II) is easily reduced to Cu(I) when subjected to XPS analysis. In turn, the formation of Cu(II) is the consequence of an oxidative thermal treatment under the air flow of the plant roots. This hypothesis is reinforced by the reaction of Eco-Cu with aqueous ammonia and the formation of deep blue [Cu(NH3)4]2+ complexes.
In the particular case of ULLMANN-type arylation reactions employing electron-poor aryl halides (Table 8, entries 4-9 – Table 9, entries 3, 10-12), did the authors perform control experiments in the absence of the catalyst? These examples could be due to pure SNAr reactions and such control experiments would be helpful in order to demonstrate that the reported reactions are unambiguously due to the presence of the catalyst.
Answer: This comment is interesting, however N- and O-arylation reactions did not work without catalyst or with CuO.
In addition, in the diaryl ether series, the authors obtained excellent results using electron-poor phenols, in particular 4-cyanophenol (Table 9, entries 10 & 12). This result is remarkable because in most cases, electron-poor phenols are poor coupling partners for such reactions. However, 4-cyanophenol has only be coupled to electron-poor 4-nitrophenyl bromide/chloride. An additional example using phenyl iodide/bromide/chloride (and/or 4-methoxyphenyl iodide/bromide/chloride) as aryl halide would be relevant here. Similarly, 4-nitrophenol would also be interesting to be tested as electron-poor phenolic partner.
Answer: We thank the reviewer for this relevant remark. However, all examples from the references [65] have been proposed.
TYPOS
- Page 4 lines 137-138: … more copper element than the plants …
Answer: This was corrected.
- Page 5 line 183: the sentence ‘Surprisingly in the … 1530 cm-1.’ is suspect
Answer: This was corrected and noticed also by the reviewer 1. Sentence: “Surprisingly in the case of Eco-Cu1 and Eco-Cu3 and 1530 cm-1” was replaced with: “Surprisingly in the case of Eco-Cu1 and Eco-Cu3 we have also observed absorption bands at 1529 and 1530 cm-1”
- Page 6 lines 194-195: … of the catalyst can be evaluated by measuring …
Answer: This was corrected and also noticed by the reviewer 1.
- Page 6 lines 225-226: the sentence ‘Therefore, … for Cu(II) ions.’ is suspect
Answer: This was corrected. The original sentence “Therefore, the proposed strategy for the rapid decomposition of parathion in aqueous media was the use the natural affinity of sulphur for Cu(II) ions.” Was replaced with: “Therefore, the proposed strategy for the rapid decomposition of parathion in aqueous media was based on the use the natural affinity of sulphur to coordinate with the Cu(II) ions.
Many thanks for taking the time to review our changes; we hope these modifications are in accord with the reviewer's and editor’s expectations.
Yours sincerely,
Claude Grison and the authors
Round 2
Reviewer 2 Report
In this revised version, C GRISON et al. have taken careful consideration of the first comments of the reviewers. As a result, I recommend publication of this manuscript with few minor changes:
- INTRODUCTION – I would suggest to add the following references in the added part devoted to heterogeneous ULLMANN-type versions = i) Green Chem 2009, 11, 1121 – Adv Synth Catal 2009, 351, 2369 – Tet Lett 2015, 56, 419 for representative examples using organic (bio)polymers as supports, ii) J Org Chem 2018, 83, 6408 for a representative example using zeolites as support
- Regarding the competition between SNAr & Cu-catalyzed ULLMANN-type coupling reactions, it would be relevant to add in the text that as mentioned in the comments to authors, ‘N- and O-arylation reactions did not work without catalyst or with CuO’. Moreover, the reference Chem Eur J 2014, 20, 5231 could be added as supporting reference.
Author Response
In this revised version, C GRISON et al. have taken careful consideration of the first comments of the reviewers. As a result, I recommend publication of this manuscript with few minor changes:
INTRODUCTION – I would suggest to add the following references in the added part devoted to heterogeneous ULLMANN-type versions = = i) Green Chem 2009, 11, 1121 – Adv Synth Catal 2009, 351, 2369 – Tet Lett 2015, 56, 419 for representative examples using organic (bio)polymers as supports, ii) J Org Chem 2018, 83, 6408 for a representative example using zeolites as support
Answer. We included the following sentence along with the requested references.
“Additionally, representative examples of Ullmann-type reaction using organic (bio)polymers,[53-55] or zeolites,[56] as supports should be mentioned.”
53. Benyahya, S.; Monnier, F.; Wong Chi Man, M.; Bied, C.; Ouazzan, F.; Taillefer, M.
Sol–gel immobilized and reusable copper catalyst for arylation of phenols from aryl bromides Green Chem., 2009,11, 1121-1123
54. Bhadra, S.; Sreedhar, B.; Ranu, B.C. Recyclable Heterogeneous Supported Copper‐Catalyzed Coupling of Thiols with Aryl Halides: Base‐Controlled Differential Arylthiolation of Bromoiodobenzenes Adv. Synth. Catal. 2009, 351, 2369 – 2378.
55. Bodhak, C.; Kundu, A.; Pramanik, A. An efficient and recyclable chitosan supported copper(II) heterogeneous catalyst for C–N cross coupling between aryl halides and aliphatic diamines Tet Lett 2015, 56, 419-424.
56. Garnier, T.; Danel, M.; Magné, V.; Pujol, A.; Bénéteau, V.; Pale, P.; Chassaing, S. Copper(I)–USY as a Ligand-Free and Recyclable Catalyst for Ullmann-Type O-, N-, S-, and C-Arylation Reactions: Scope and Application to Total SynthesisJ. Org. Chem., 2018,83, 6408–6422
Regarding the competition between SNAr & Cu-catalyzed ULLMANN-type coupling reactions, it would be relevant to add in the text that as mentioned in the comments to authors, ‘N- and O-arylation reactions did not work without catalyst or with CuO’. Moreover, the reference Chem Eur J 2014, 20, 5231 could be added as supporting reference.
Answer: The following sentence was added just below paragraph 4.3. and reference as number 96
“Importantly N- and O-arylation reactions did not work without Eco-Cu catalyst or with CuO [96].”
96. Drapeau, M.P.; Ollevier, T.; Taillefer, M. On the Frontier Between Nucleophilic Aromatic Substitution and Catalysis Chem Eur J 2014, 20, 5231-5236.
